# A Phosphonic Functionalized Biopolymer for the Sorption of Lanthanum (III) and Application in the Recovery of Rare Earth Elements

Mohammed F. Hamza [1,2,*], Walid M. Abdellah [2], Doaa I. Zaki [2], Yuezhou Wei [1,3,*], Khalid Althumayri [4], Witold Brostow [5,6] and Nora A. Hamad [5,6,7]

1 School of Nuclear Science and Technology, University of South China, Hengyang 421001, China
2 Nuclear Materials Authority, El-Maadi, Cairo 11728, Egypt
3 School of Nuclear Science and Engineering, Shanghai Jiao Tong University, Shanghai 200240, China
4 Department of Chemistry, College of Science, Taibah University, Al-Madinah Al-Munawarah 30002, Saudi Arabia
5 Laboratory of Advanced Polymers & Optimized Materials (LAPOM), Department of Materials Science and Engineering, University of North Texas, 3940 North Elm Street, Denton, TX 76207, USA
6 Department of Physics, University of North Texas, 3940 North Elm Street, Denton, TX 76207, USA
7 Chemistry Department, Faculty of Science, Menofia University, Shebin El-Kom 32511, Egypt
* Correspondence: m_fouda21@usc.edu.cn (M.F.H.); yzwei@usc.edu.cn (Y.W.);
Tel.: +20-111-668-1228 (M.F.H.); +86-771-322-4990 (Y.W.)

**Abstract:** Phosphonic acid functionalization of gellan gum and chitosan biopolymers was successfully performed. In the first step, the sorption was investigated using La(III) ions before testing for the recovery of rare earth elements (REEs) from pretreated industrial acidic leachate. The sorbent was characterized by Fourier-transform infrared (FTIR), scanning electron microscopy with energy dispersive X-ray analysis (SEM-EDX), thermogravimetric analysis (TGA), Brunauer–Emmett–Teller (BET), and pH of zero charge ($pH_{PZC}$) determination. FTIR and EDX results show efficient grafting of phosphoryl groups. The sorption was determined for the crude materials before functionalization (PGEG) and after phosphorylation (TBP-PGEG). More efficient sorption was seen for phosphorylated sorbent than for the crude composite. The sorption capacity is 0.226 mmol La $g^{-1}$ for the PGEG while the value is 0.78 mmol La $g^{-1}$ for the TBP-PGEG. We infer that phosphonate groups participate in the sorption. The most effective sorption is at pH = 4. The kinetic behavior was described using pseudo first-order equations (PFORE), pseudo second-order equations (PSORE), and resistance to intraparticle diffusion (RIDE). The sorption isotherms can be better represented by Langmuir and Sips equations than by the Freundlich equation. The sorbent shows high stability performance during reuse cycles with a limit on the decrease in the sorption performances and stability in the desorption performances. We have thus developed a good tool for the recovery of REEs with a selectivity higher than that of the non-functionalized components.

**Keywords:** biopolymer; phosphorylated sorbent; uptake kinetics; sorption isotherms; rare earth elements

## 1. Introduction

The recycling of materials and thus sustainability are increasingly important [1]. This applies also to the recovery of rare earth elements (REEs) [2]. Lanthanum is a light rare earth element with an atomic number of 57. It is found abundantly in the environment; it is a soft, malleable, ductile, and silver-white metal. Lanthanum is naturally found in sediments; it oxidizes rapidly in air and reacts with water to form the hydroxide. The salts of lanthanum are insoluble in water. It is used in the field of optics as lenses, the production of catalysts, the polishing of glass, and also in radiation absorbing glasses [3]. Like most other rare earth elements, the usual oxidation state is +3. It usually occurs together with

cerium and the other rare earth elements. It mainly occurs in monazite (Ce, La, Y, Th) $PO_4$ and bastnäsite (La, Ce)$FCO_3$ ores in measurable quantities [4].

Several procedures were used for the extraction of REEs from the secondary resources such as phosphate rocks [5], minerals with low-grade concentration [6], and wastes from industrial materials [7–9])). Hydrometallurgical techniques have been used for dissolving metal ions from solid materials to leachate solution [10–18]. Leachates are contaminated liquids generated from water percolating through a solid waste disposal site, accumulating contaminants, and moving into subsurface areas. Target metal ions have been obtained by solvent extraction or direct precipitation [19–25]. Both these methods were used for high-grade metal ions and are inefficient for low-concentration solutions. Some functional groups play a vital role in the extraction process, most of them based on phosphorus compounds such as alkylated phosphoric acid [26–28], phosphonate groups [3,20,29–32], extractants containing organo-phosphonic groups [33–36], ionic liquids with phosphonium [37,38], or alkylated phosphine oxide [39,40].

For the low-grade concentrations, precipitation and solvent treatment are not applicable; hence, the target metal ions are extracted through other processes. These techniques involve using an impregnated sorbent, for instance, in REEs for the enrichment and separation from acidic solutions. This technique is fast and has a high capacity [41–44]. Small-size nanoscale sorbents were used for the mitigation of mass transfer [45]. The incorporation of magnetite nano scale particles into polymer improves the kinetics and separation efficiency [46–48]. Other conventional adsorbents including chelating [41,49–51] and ion-exchange sorbents [52–56] have been used for recovering target metals from acidic and alkaline solutions. These materials have functional groups for binding with metal ions even present at low concentrations with high selectivity. Such sorbents include Purolite (C-100) [56] and Dowex (50W X8) [53]. Multi-functional sorbents with sulfonic, carboxylic, and diphosphonic groups [57] constitute the Purolite family [50].

High affinity of the phosphorus extractants towards REEs and fast process kinetics provided a basis for designing a number of solid phase adsorbents such as Tulsion (CH-96) and (T-PAR) resins [58,59], Tulsion (CH-93) [33], and Diphonix [60].

Chitosan particles are obtained from deacetylated chitin. Chitin appears naturally in the crustaceans such as crab, crayfish, shrimp shells, and cuttlefish, in insect cuticles, and cell walls of the fungi [61]. Chitin is well known because of its uses in water treatment (removal of pollution such as dyes or heavy metals). The abundance of hydroxyl and amine groups in its structures [62,63] makes this biopolymer a vital component in many adsorbents—with possibilities of further modification (chemically) by grafting of additional groups for increasing the capacity and selectivity, also by designing (physically) beads and nanomaterials for improving the kinetic properties. Moreover, the presence of such groups enables the reinforcement of the final products by crosslinking by binding with amines and hydroxyls [64–66].

Several studies were performed for metal recovery using modified chitosan particles. Crosslinked lanthanum-loaded chitosan/silica gel particles was performed through designing a $LaCl_3 \cdot 7H_2O$, chitosan and silica gel composite—which crosslinked using glutaraldehyde crosslinker yielding (LaSiCS), an excellent sorbent for chromium removal [67]. Another sorbent was synthesized and explored for the removal of lanthanum and lead ions from water using a manganese dioxide chitosan composite (crosslinked by formaldehyde). The sorbent shows a high affinity and rapid reaction with these ions [68]. Sorbents based on chitosan and polyvinylpyrrolidone (PVP) particles were designed for the separation of La(III) from (La(III), Ce(III), and Sm(III)). Tartaric acid with a concentration of 0.25 M was used as an eluent for lanthanide from Ln(III)IP-CS/PVP particles with a selective extraction capabilities [69].

In this work, the chitosan particles were coupled with gellan gum through crosslinking reagents (EPI and GA); functionalization was performed by phosphorylation through tributyl phosphate (TBP) and phosphoric acid ($H_3PO_4$). Both sorbents were characterized by FTIR, TGA, BET, pHpzc, and SEM-EDX analyses. The second part of this work involved

the determination of the sorption properties of sorbents toward La ions—including effects of pH, uptake kinetics, sorption isotherms (the sorption efficiency of the functionalized sorbent exceeding by more than three folds compared to the nonfunctionalized composite, with fast kinetic properties (25 min compared to 40 min)), and sorption from polymetallic equal molar solutions (with respect to the pH of the loaded solution). Finally, the last part of the study involved the recovery of REEs from an acidic leachate solution for the evaluation of sorbents toward natural solutions, which shows a high affinity toward the REE solution.

## 2. Materials and Methods

### 2.1. Materials

Gellan gum was supplied by Gino Biotech—Zhengzhou, Henan, China. Chitosan (medium molecular weight), glutaraldehyde ((GA)50%, $w/w$ in water), and poly(vinyl alcohol) (PVA) with molecular wt. between 9000 and 10,000 (80% hydrolyzed) were supplied by Sigma-Aldrich (Shanghai) Trading Co. Ltd., Pudong, Shanghai, China. Tributyl phosphate ((TBP); $CH_3(CH_2)_3O)_3PO$, >99%, epichlorohydrin (EPI) $C_3H_5ClO$, 99% and phosphoric acid $H_3PO_4$, 99.99% were purchased from Shanghai-Makclin- Biochemical Co., Ltd. (Shanghai, China). Terbium (III) sulfate (used in the selectivity test) and lanthanum (III) sulfate were supplied by the Rare Earth (RE) Metallurgy and Functional Materials Co., Ltd., China (National Engineering Research Centre). Silicon (source of silica standard solution for selectivity experiments) with initial concentration of 1000 ppm was supplied by Guobiao-Inspection and Certification-Co. Ltd. Huairou-District, Beijing, China. The remaining reagents used in this study were Prolabo products, VWR, Radnor, PA, USA.

### 2.2. Synthesis Procedures

Phosphorylating agent was synthesized by addition of 25 g of TBP solution in a three-nicked round bottom flask—followed by addition of 8 mL phosphoric acid solution. The reactor was closed and the reaction mixture was refluxed at 80 °C for 1 h. After cooling, another addition to the mixture was 8 g of EPI; the solution was refluxed again for 4 h at 90 °C, then left for cooling (named as part A); see Scheme 1.

**Scheme 1.** Synthesis pathway of the phosphorylating agent.

A mixture of 1.2 g of gellan gum and 1.2 g of chitosan was dissolved in 30 mL of 5% acetic acid solution, then 10 mL of 2% PVA added under continued stirring until homogeneity was achieved. Then, EPI was added as crosslinker and the system left in an oven at 40 °C overnight. The material so produced is gelatinous (named as part B).

We have also made a non-functionalized material (1)—used in a comparable study of the sorption with the phosphorylated composite. The same procedure was repeated for B and continued by addition of EPI, then refluxed at 60 °C; the material was poured into a 500 mL solution of GA (15%) and stirred overnight before being filtered and dried to produce a precipitate (PGEG).

Scheme 2 shows the synthesis procedure of PGEG.

Addition of B to A was followed by refluxing for 12 h at 90 °C. The final product was obtained by pouring the mixture into a beaker containing 5% NaOH with 20 mL GA solution to provide a phosphorylated composite (Scheme 3). The mixture was stirred for 10 h at room temperature and the precipitated product was filtered and washed by water and acetone for removal of the unreacted substrates.

**Scheme 2.** Synthesis procedures of PGEG composite.

**Scheme 3.** Scheme of the phosphorylation method of PGEG (TBP-PGEG).

## 2.3. Characterization of Materials

FT-IR spectroscopy was performed using the IRTracer-100 (FT-IR spectrometer Shimadzu, Tokyo, Japan). Samples were dried at 65 °C for 10 h and ground with KBr (dried at 70 °C for 12 h) before creation of each disk for analysis. The $pH_{PZC}$ values of both functionalized (TBP-PGEG) and non-functionalized (PGEG) products were determined with 0.1 M NaCl solution using the drift method [70]; the pH in this study ranged from 1 to 11 after keeping the samples for 48 h. The pore size and pore volume were determined using the nitrogen sorption desorption isotherms (Brunauer-Emmett-Teller or BET analysis). Both samples were treated with nitrogen gas for 5 h at 130 °C. SEM and EDX analyses were performed using Phenom, ProX-SEM, Thermo Fisher Scientific, Eindhoven, the Netherlands. Thermogravimetric analysis (TGA) was performed using Netzsch-STA, 449 F3-Jupiter, Netzsch Gerätebau-H Gmbh, Selb, Germany, performed in nitrogen atmosphere with the temperature increase rate of 10 °C min$^{-1}$. The pH of solutions was adjusted before sorption and measured after loading; this was also the case for the $pH_{PZC}$ using a Compact (pH-S220-Seven/Ionometer) from Mettler-Toledo, Shanghai, China. The liquid samples including metal ions went through a filtrate membrane (1.2 μm) before testing for metals using the Inductively Coupled Plasma (ICP) analysis (ICPS; 7510-Shimadzu, Tokyo, Japan).

*2.4. Metal Sorption Properties*

The batch process was used in the sorption from either synthetic or natural leachate solutions. An amount of dry sorbent (m, g) was added to a volume of solution (V, L) that had a specific concentration of metal ions with initial concentration ($C_0$, mmol $L^{-1}$) at a certain pH value (resulting from the pH study). The shaking speed during the sorption process is approximately 170 rpm at the temperature T = 22 ± 2 °C. At the end of sorption, the samples were collected, filtered, and the metal content determined using the ICP equipment ($C_{eq}$, mmol $L^{-1}$). The capacity of the matrix ($q_{eq}$, mmol $g^{-1}$) was calculated using the following equation:

$$q_{eq} = (C_0 - C_{eq}) \times V/m \tag{1}$$

The sorbent concentration was determined in a solution with an equimolar concentration before application of a leaching solution containing REEs. The samples from kinetic experiments were used for kinetic elution through treatment with 0.5 M $HNO_3$. The sorption recycling was performed up to 5 cycles (water rinsing was systematically perfumed between each cycle), in which the sorption capacity and efficiency were determined for investigate the loss %. Tables S1 and S2 summarize the kinetic and isotherm sorption parameters, respectively; the kinetics were performed using PFORE, PSORE, and RIDE (pseudo first-/second-order rate equations, and resistance to intraparticle diffusion, respectively), while the sorption isotherms were fitted using Freundlich, Langmuir, and Sips equations. The fitting quality were determined using $R^2$ (determination coefficient, also called squared multiple correlation coefficient) and AIC (the Akaike Information Criterion) [71]. The applied natural liquor was collected after pretreated acidic leaching of ore material after treatment for U recovery. The original leaching liquor with 410 mg REE $L^{-1}$ and 530 mg U $L^{-1}$ as appeared in Table S3 was firstly treated with amberlite IRA 400; the uranium decreased to 38 mg U $L^{-1}$ with efficiency (sorption) around 93%. However, the REE(III) decreased to 380 mg $L^{-1}$ with efficiency up to 7.3%. The other elements were reported in Table 1

**Table 1.** Constituents of solutions after treatment for uranium (VI) removal.

| Constituent | Conc. (mg/L) | Constituent | Conc. (mg/L) |
|---|---|---|---|
| U | 38 | $Al_2O_3$ | 1896 |
| REE | 380 | Pb | 16.2 |
| Fe | 1852 | Zr | 15 |

## 3. Results

*3.1. Sorbent Characterization*

3.1.1. Textural Properties (BET Analysis and Morphological (SEM) Observations)

The composite after functionalization (TBP-PGEG) shows an increase of the specific surface area (SSA) equal to 22.15 $m^2$ $g^{-1}$ when compared to 17 $m^2$ $g^{-1}$ for the PGEG. The porous volume is higher as well, namely, 0.014 $cm^3$ $g^{-1}$ for PGEG, while it is 0.068 $cm^3$ $g^{-1}$ for TBP-PGEG. This comparison results from the isotherms of adsorption–desorption. The cumulative pores volume was increased by functionalization from 0.049 to 0.058 $cm^3$ $g^{-1}$.

3.1.2. SEM-EDX and Elemental Analysis

The SEM analysis shows an irregular shape with small pores of sorbents. Figure 1a,b shows the SEM morphological structure of both sorbents—smaller pores in functionalized sorbent than in the pristine composite. Figure 1c,d shows the EDX analysis of both sorbents; an increase of O contents from 45.5 to 46.31% and the appearance of 3.11% P in the functionalized sorbent confirm the successful modification on the PGEG composite.

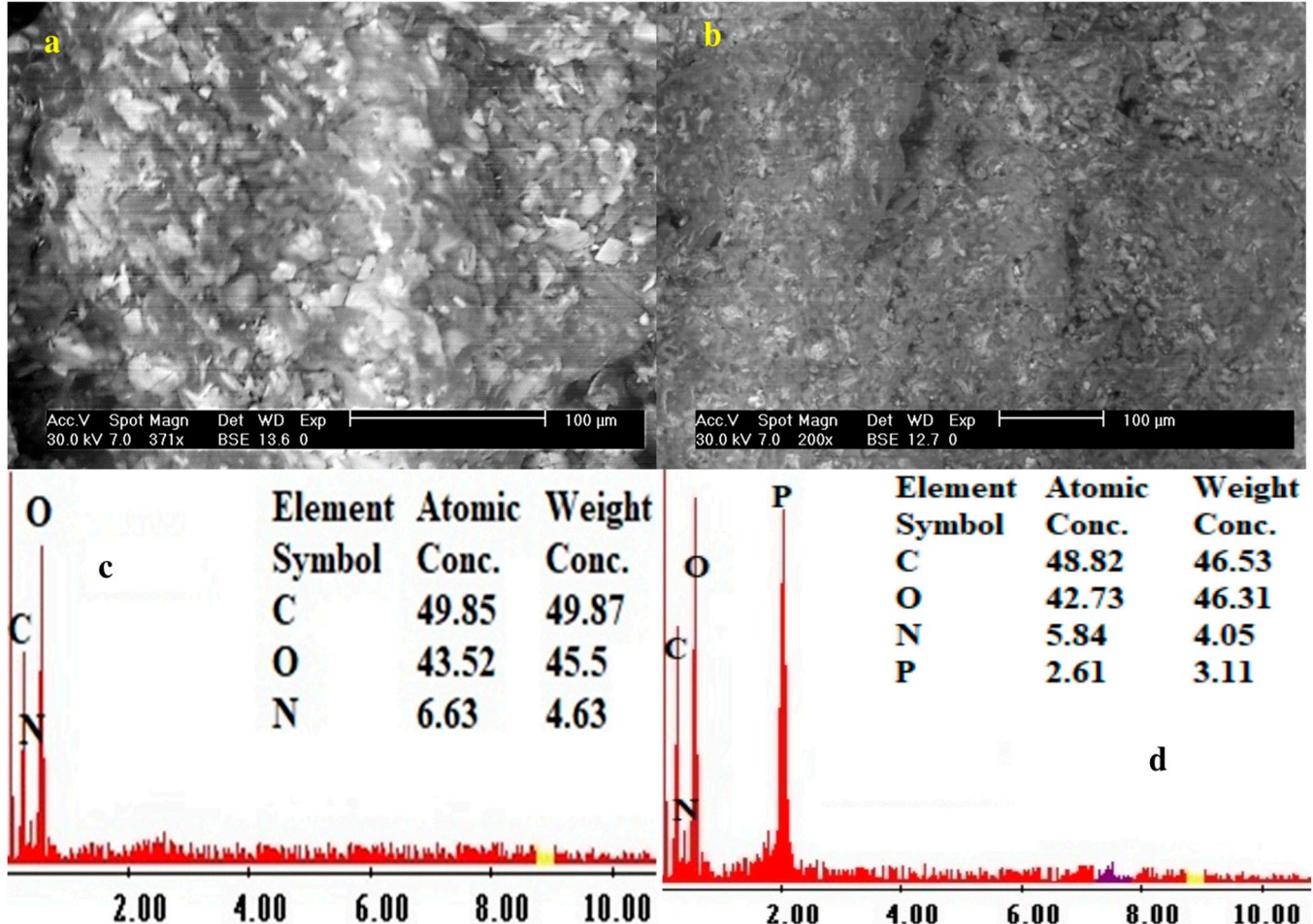

**Figure 1.** The surface morphological structure (SEM) and chemical composition (EDX) of PGEG (**a**,**c**) and TBP-PGEG (**b**,**d**) sorbents.

These results are confirmed by the elemental analysis of both PGEG and TBP-PGEG; see Table 2. As already noted, efficient phosphorylation is seen in an increase of the O% concentration and in the appearance of P in the functionalized sorbent. The O% concentration increases only a little, from 40.68 to 41.72% for the PGEG and TBP-PGEG, respectively, while the P appears with 4.83%.

**Table 2.** The elemental analysis results for PGEG and TBP-PGEG composites.

| Sorbent | C [%] | N [%] | H [%] | O [%] | P [%] |
|---|---|---|---|---|---|
| PGEG (%) | 47.56 | 4.78 | 6.98 | 40.68 | 0 |
| PGEG (mmol) | 39.60 | 3.41 | 69.26 | 25.43 | 0 |
| TBP-PGEG (%) | 42.14 | 4.13 | 7.18 | 41.72 | 4.83 |
| TBP-PGEG (mmol) | 35.09 | 2.95 | 71.24 | 26.08 | 1.56 |

### 3.1.3. Thermal Decomposition Analysis

Both sorbents show very similar thermal decomposition profiles in the TGA, with the percentage weight loss and the thermal stability shown in Figure 2. The weight loss of the sorbent values is 97.77 and 88.15% for PGEG and TBP-PGEG, respectively. The hydrocarbon concentration increases accordingly. Three loss profiles are seen for the sorbents—as discussed below.

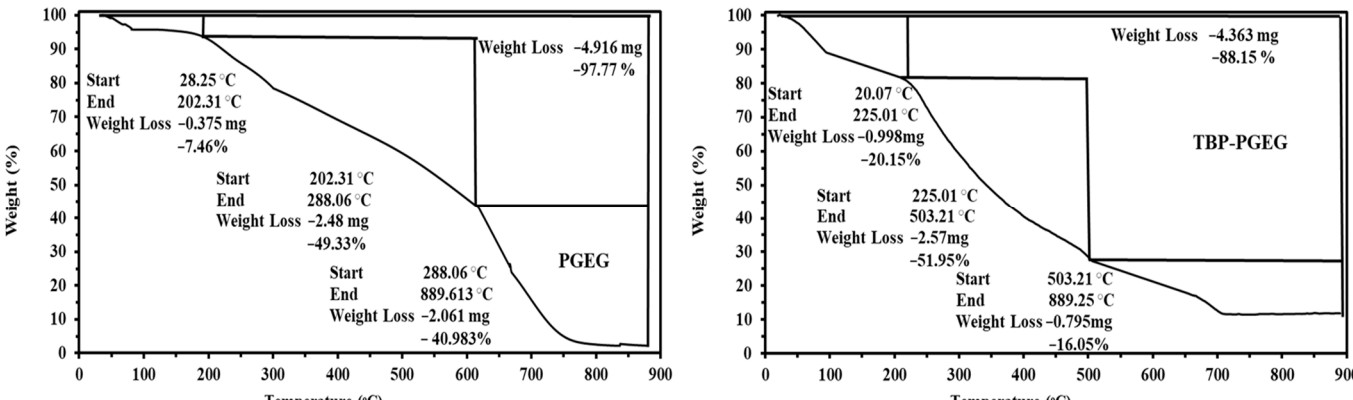

**Figure 2.** Thermogravimetric (TGA) analysis of the PGEG and TBP-PGEG sorbents.

The first stage is seen at 202 and 225 °C for PGEG and TBP-PGEG, respectively; this stage is due to the water loss (7.46 and 20.15 %, respectively) [72,73].

The second loss stage is seen below 288 and 503 °C for both sorbents, respectively. This stage is due to degradation of amines, cleavage of the crosslinking linkages, and decomposition of the network frames [74]. At this stage, there is a further weight loss, by 49 and 51% for both sorbents, respectively, followed by the final loss around 41 and 16%, respectively—due to depolymerization of the remaining hydrocarbon and to the char formation.

The Differential Thermogravimetric Analysis (DrTG) (Figure S1) for PGEG shows a series of peaks around 66, 139, 219, and 761 °C and two peaks for TBP-PGEG around 411 and 785 °C. We conclude that the phosphorylation reaction results in the successful grafting of the phosphonic groups and in achieving higher thermal stability.

### 3.1.4. FTIR Spectroscopy

Figure 3a,b shows a comparison of both sorbents before and after functionalization. We see the results for the sorbent after loading with La(III) and after five cycles of sorption and desorption.

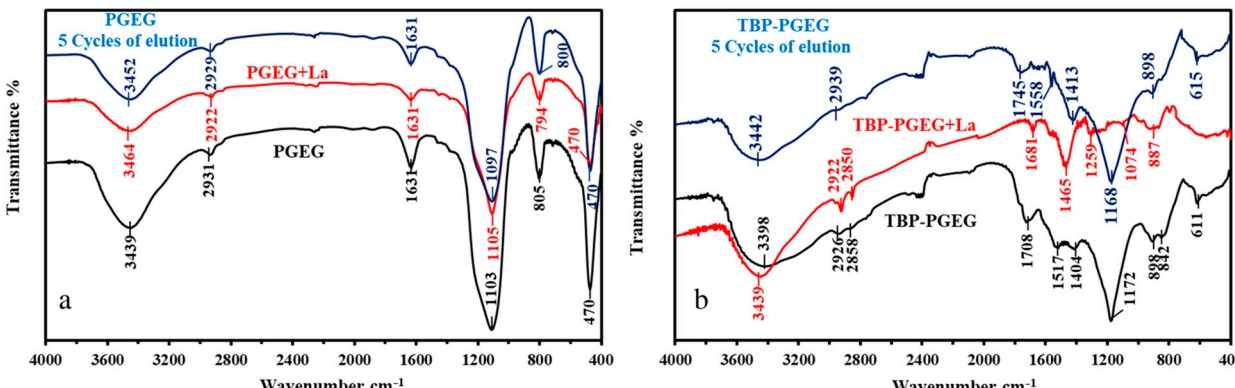

**Figure 3.** FTIR analysis of PGEG (**a**) and TPB-PGEG (**b**), after loading with La and after 5 cycles of sorption desorption.

The FTIR results confirm the structure of PGEG and TBP-PGEG sorbents. Both sorbents show peaks of OH and NH. Extra peaks are seen after phosphorylation reactions due to a change of the environment of these groups as reported in Table S4.

Briefly, we see a displacement of these peaks and changes in the peaks shape [75,76]. In addition, some new peaks are seen for the grafted moieties [77–82]; a peak at 1708 cm$^{-1}$ and 1404 cm$^{-1}$ for C=O and P=O, respectively [83] with a displacement of the NH peak

from 1630 cm$^{-1}$ to 1517 cm$^{-1}$; a new peak at 1172 cm$^{-1}$ for the P–O–C; a change in the shape of the peak at 805 cm$^{-1}$ now divided into two parts at 842 cm$^{-1}$ and 898 cm$^{-1}$ for P–C groups [81,83]; and a change in the width of the peak at 611 cm$^{-1}$ for the TBP-PGEG (the overlapping of P–O and Fe–O peaks) [84]. After metal sorption, some peaks, especially for OH, NH, and C=O, disappeared or shifted—apparently, because of the metal sorption [84–87]. The peaks in the 2800–1950 cm$^{-1}$ range [88,89] correspond to the asymmetric aliphatic C–H. The peak at 1631 cm$^{-1}$ in the pristine sorbent becomes split into two parts around 1700–1750 cm$^{-1}$—an effect of phosphorylation on the C=O environment while another peak appears at 1510–1580 cm$^{-1}$ for the NH amine groups [90]. Phosphorylation is followed by peak broadening in the region of 800–1600 cm$^{-1}$—mainly due to the NH, C=O, P–O–C, P=O, P–C, and COO$^{-}$ [90]. The changes after grafting and the shifts resulting from metal loading are discussed in Table S4, while Figure S2 shows a comparison of the spectra.

### 3.1.5. pH$_{PZC}$ Analysis

Another evidence of the modifications on the PGEG surface including the presence of the solvent is the change of the pH with respect to pH$_0$. Figure S3 shows a comparison of the pH$_{PZC}$ for PGEG and after functionalization. We see that the pH shifts toward higher values. The original components in the PGEG are amines (from chitosan) with pKa = 4.5 for the primary amine, 6.7 for the secondary amine, and 11.6 for the tertiary amine. Clearly, the acid base characteristics is affected by the phosphorylation progress. Kołodyńska and her colleagues [91] studied the progress of sorption in a series of resins with lanthanum; they reported pH$_{PZC}$ values in the range from 8.13 to 9.93 for the aminophosphonic resins.

### 3.2. Detection in Synthetic Solutions

### 3.2.1. Effect of pH

Figure 4a–c shows a comparable study of La(III) sorption using PGEG and TBP-PGEG sorbents. Figure 4a shows the effect of pH on sorption at a pH$_0$ ranging from 1 to 6. The sorbents provide similar profiles, with a low sorption capacity at lower pH values then increasing gradually until saturation. For TBP-PGEG at lower pH values, the functional groups are protonated; these positively charged groups repulse the positively charged metal ions. The average loading capacity after saturation is 0.226 mmol La g$^{-1}$ and 0.78 mmol La g$^{-1}$ for PGEG and TBP-PGEG, respectively. This reflects the successful grafting of phosphonate groups; the loading capacity is higher than that of the pristine sorbent by ≈3.5 times. TBP-PGEG has high capacity in the entire pH range; both sorbents show the saturation equilibrium at pH = 4.

Figure 4b shows the variation of the pH through La(III) sorption. The pH changes by 0.3 units from the initial concentration and a little higher especially for the high pH values. We show in Figure 4c the plot of log$_{10}$ of the D (distribution ratio = q$_{eq}$/C$_{eq}$) vs. pH$_{eq}$. We recall that q$_{eq}$ has been defined in Equation (1). We see in Figure 4c that the plot is approximately linear, with the slope ≈ +0.49. Apparently, the metal ions are bound to the sorbent through chelation. The sorbent is partially deprotonated and the binding occurs through the lone pairs of electrons localized on atoms.

Figure S4 shows relative concentrations of La containing components as functions of pH. LaSO$_4^{+}$ has the highest concentration below pH = 4, coexisting in that pH range with the free cationic species La$^{3+}$. The anionic species in disulfate form La(SO$_4$)$_2^{-}$ is seen at pH < 3. The sorbents are partially protonated; the pH$_{PZC}$ is ≈6.17 and ≈6.64 for PGEG and TBP-PGEG, respectively; this is evidence for binding free electrons to the positively charged metal ions—what explains the decreasing repulsion between functional groups and metal ions. The binding is achieved through OH, NH, COOH, and phosphonic groups—providing high loading capacity. This is verified by a decrease and displacement of these peaks after sorption; see again the FTIR spectra.

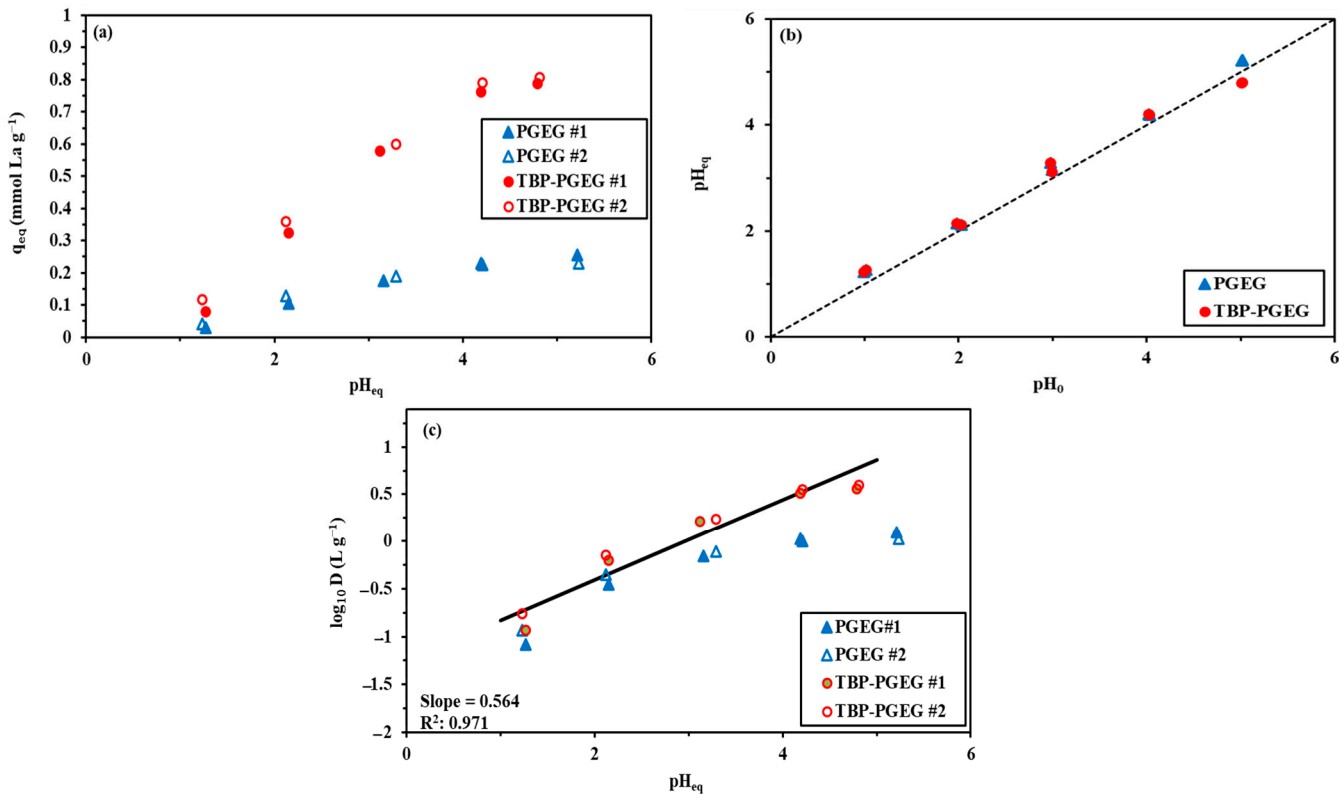

**Figure 4.** Effects of pH on sorption capacity of La (**a**), pH variation (**b**) and plot of $\log_{10}D$ against $pH_{eq}$ (**c**).

Figure 5 shows the EDX diagrams of the loaded sorbents. The S element reflects the sorption of the sulfate species; the presence of P after grafting verifies the successful phosphorylation reaction while the high contents of La in TBP-PGEG shows a high loading capacity of the phosphonate sorbent compared to the crude one.

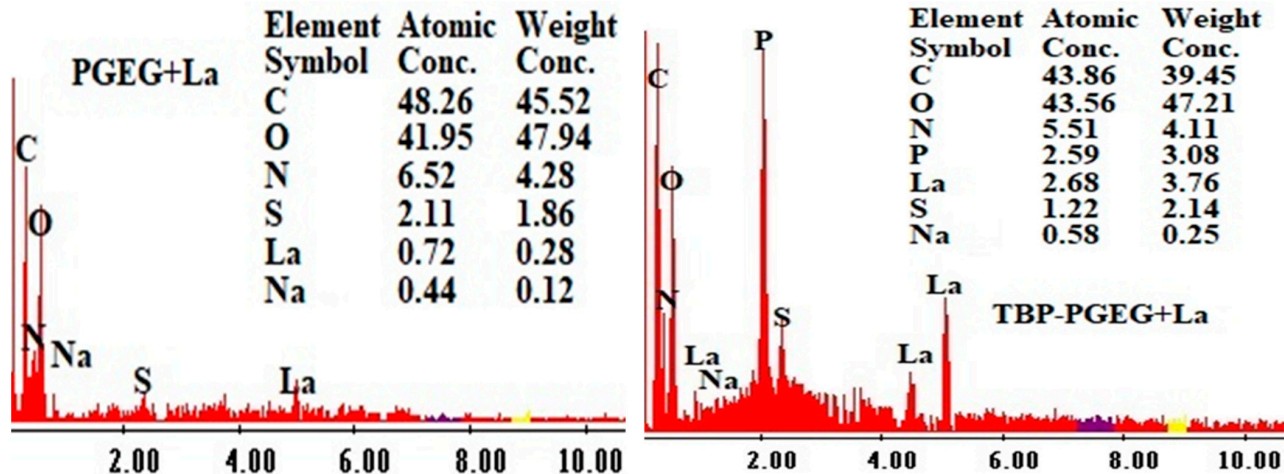

**Figure 5.** EDX analysis of PGEG and TBP-PGEG after La sorption.

### 3.2.2. Uptake Kinetics

The uptake kinetics depends on a variety of factors, including resistance to diffusion—which is different in film than in bulk, and there is also intraparticle diffusion. The homogeneity distribution of the sorbent particles depends on the agitation velocity and also on the film and bulk diffusion. The uptake kinetics was investigated considering PFORE,

PSORE, and RIDE (using the Crank equation). PFORE (usually reflecting the physical adsorption) provides a very good fit seen in Figure 6, while Figure S5—left shows the experimental and PSORE results related to chemical adsorption and Figure S5—right shows the experimental and RIDE results. Goodness of the fit of the $R^2$ and AIC values are provided in Table 3. The equilibrium was achieved at 40 and 25 min for PGEG and TBP-PGEG, respectively. Another advantage of grafting phosphoryl groups is the accelerated kinetics of sorption. This is related to the high affinity of phosphonate groups to REEs [92].

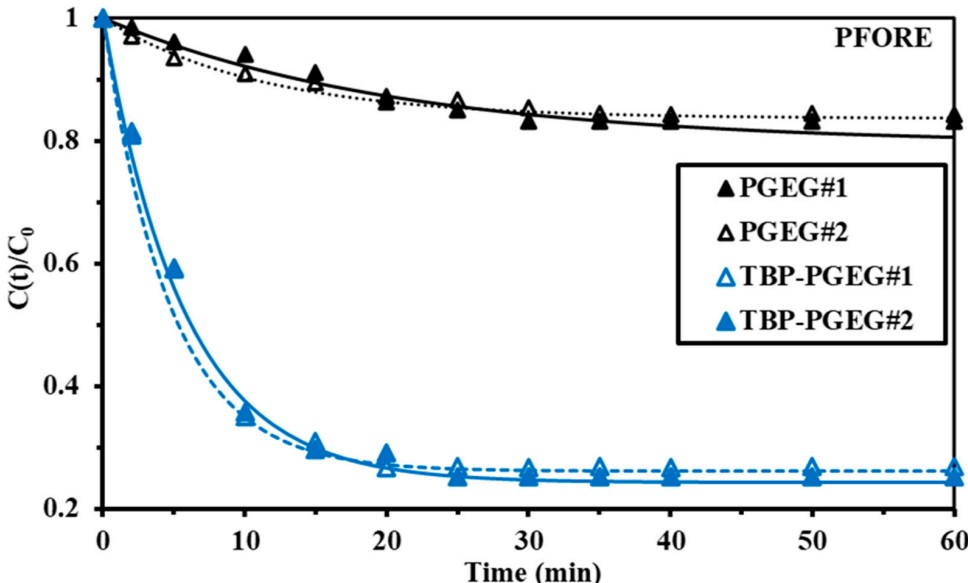

**Figure 6.** Kinetic profile of the fitted equation (PFORE) of the experimental data for La sorption of PGEG and TBP-PGEG sorbents at $pH_0 = 5$.

**Table 3.** Modeling of La(III) uptake kinetics for PGEG and TBP-PGEG sorbents at $pH_0 = 5$.

| Model | Sorbent | PGEG | | TBP-PGEG | |
|---|---|---|---|---|---|
| | Parameter | 1 | 2 | 1 | 2 |
| | $q_{eq,exp.}$ | 0.191 | 0.184 | 0.793 | 0.785 |
| PFORE | $q_{eq,1}$ | 0.186 | 0.177 | 0.789 | 0.775 |
| | $k_1 \times 10^2$ | 3.273 | 3.443 | 4.54 | 4.73 |
| | $R^2$ | 0.995 | 0.985 | 0.983 | 0.989 |
| | AIC | −125.1 | −132.1 | −133.7 | −145.3 |
| PSORE | $q_{eq,2}$ | 0.218 | 0.223 | 0.947 | 0.856 |
| | $k_2 \times 10^2$ | 2.01 | 1.97 | 2.847 | 2.11 |
| | $R^2$ | 0.921 | 0.913 | 0.935 | 0.927 |
| | AIC | −21 | −31 | −39 | −38 |
| RIDE | $D_e \times 10^{13}$ | 1.36 | 1.49 | 2.06 | 2.25 |
| | $R^2$ | 0.932 | 0.941 | 0.921 | 0.923 |
| | AIC | −41 | −44 | −54 | −57 |

Units: $q_{eq}$: mmol g$^{-1}$; $k_1$: min$^{-1}$; $k_2$: L mmol$^{-1}$ min$^{-1}$; $D_e$: m$^2$ min$^{-1}$.

### 3.2.3. Sorption Isotherms

The sorption isotherms show among others the maximum sorption capacity and the affinity toward the target metal ions. Different equations were used for representing the isotherms: (a) Langmuir was used for monolayer system without chemical interactions of the sorbed molecules—that is, for homogeneous sorption, good results are seen in Figure 7; (b) Freundlich, known to be used for multi-layer-sorption, a power-type equation, provides a worse fit than the Langmuir equation; and (c) Sips, which is a combination of

Freundlich and Langmuir equations with an additional third parameter, provides a good fit—as also seen in Figure 7. It seems noteworthy that the slope of the TBP-PGEG curve steeply increases before saturation—while this is not the case for PGEG. The total sorption of PGEG is $\approx 0.45$ mmol La g$^{-1}$ while it is $\approx 1.47$ mmol La g$^{-1}$ for the TBP-PGEG (a strong increase after phosphorylation).

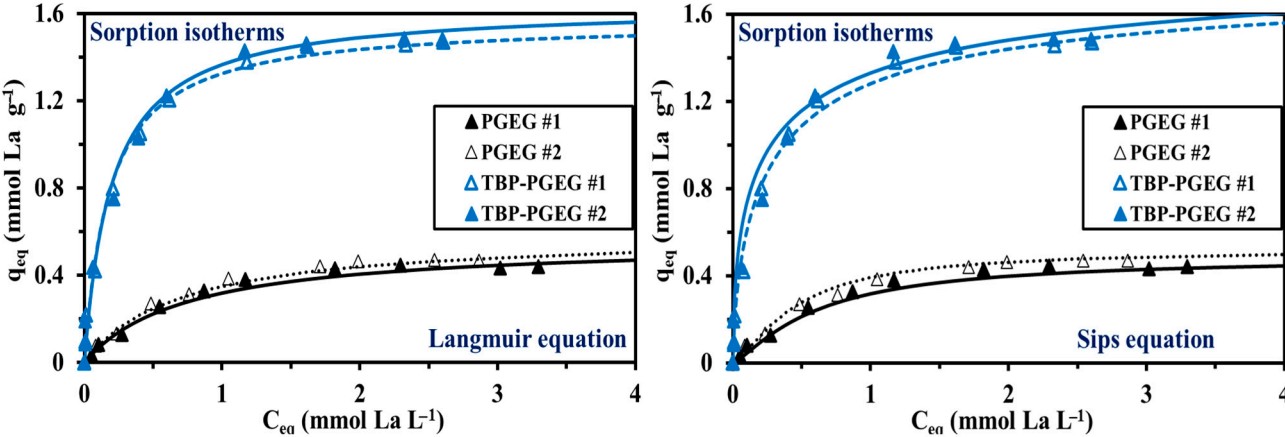

**Figure 7.** Experimental results and values calculated from Langmuir and Sips equations for the sorption isotherms of La(III) ions on PGEG and TBP-PGEG sorbents.

Figure S6 shows the bad-fitting profile (Freundlich) for both sorbents at pH$_0$ 4. It is noteworthy that the slope of TBP-PGEG is steeply increased before saturation, while the PGEG profiles show a smooth increasing of the sorption under the same conditions. The total sorption of PGEG is around 0.45 mmol La g$^{-1}$ comparing to 1.47 mmol La g$^{-1}$ for the TBP-PGEG, which confirmed a strong increasing in the sorption capacities (increasing of three-fold after phosphorylation).

Table 4 reports the model's parameters with the statistical indicators of AIC and $R^2$. The high affinity of TBP-PGEG toward REEs can be related to the Pearson's rules, also called Hard&Soft Acid–Base Theory (HSAB) [93,94]. The phosphonate groups are classified as hard bases [95], which are attracted to the hard acids such as REEs. Series of phosphonate grafted sorbents were investigated toward different metal ions in which the hydrogen bond should be yield to establish the optimization of the binding properties [96].

**Table 4.** Modeling of La(III) sorption isotherms for PGEG and TBP-PGEG sorbents at pH$_0$ = 4.

| Model | Sorbent | | PGEG | | TBP-PGEG | |
|---|---|---|---|---|---|---|
| | Parameter | Run | 1 | 2 | 1 | 2 |
| Experimental | $q_{m,exp.}$ | | 0.433 | 0.471 | 1.457 | 1.483 |
| Langmuir | $q_{m,L}$ | | 0.439 | 0.477 | 1.475 | 1.489 |
| | $b_L$ | | 4.1 | 4.4 | 5.75 | 5.43 |
| | $R^2$ | | 0.982 | 0.987 | 0.994 | 0.991 |
| | AIC | | −54 | −57 | −77 | −78 |
| Freundlich | $k_F$ | | 0.457 | 0.473 | 1.53 | 1.62 |
| | $n_F$ | | 2.18 | 2.59 | 3.88 | 4.17 |
| | $R^2$ | | 0.911 | 0.884 | 0.946 | 0.915 |
| | AIC | | −21 | −18 | −24 | −28 |
| Sips | $q_{m,S}$ | | 0.435 | 0.463 | 1.461 | 1.487 |
| | $b_S$ | | 6.85 | 5.94 | 2.18 | 1.97 |
| | $n_S$ | | 1.04 | 1.12 | 1.17 | 1.13 |
| | $R^2$ | | 0.982 | 0.989 | 0.993 | 0.997 |
| | AIC | | −60 | −53 | −67 | −69 |

Units: $q_m$: mmol g$^{-1}$; $b_i$: L mmol$^{-1}$; $n_{F/S}$: dimensionless; $k_F$: L$^{1/n_F}$ mmol$^{1-1/n_F}$ g$^{-1}$.

Table 5 reports the sorption performances of alternative sorbents for the extraction of La(III) and comparison with the results from using PGEG or TBP-PGEG sorbents. The difference in the conditions makes such a comparison only an approximation. However, we see meaningful trends through comparison of the sorption capacities, $b_L$, and equilibrium times. The TBP-PGEG is characterized by fast kinetics and highly sorption properties compared to the PGEG and most sorbents known from the literature. Functionalized-chitosan [97], alginate magnetite beads [98], MOF functionalized sorbent [99], and POH-$A_L$PEI [100] show high sorption capacities while TBP-PGEG positively distinguishes itself by its fast kinetics.

**Table 5.** La(III) sorption for alternative sorbents.

| Sorbent | Time | $pH_0$ | $q_{m,L}$ | $b_L$ | Ref. |
|---|---|---|---|---|---|
| *Mycobacterium-smegmatis* | 180 | 1.5 | 0.024 | 4.1 | [101] |
| *Citrus reticulata* peel | 60 | 5 | 1.11 | 10.6 | [102] |
| Lewatit TP-207 | 60 | 3.6 | 0.867 | 47.0 | [103] |
| Functionalized membrane of calixarene | 1440 | 5 | 1.12 | 144 | [104] |
| Functionalized chitosan | 240 | 4 | 2.03 | 0.06 | [97] |
| SQS-6 cationic sorbent in $H_3PO_4$ | 10 | 4 | 0.086 | 3.66 | [105] |
| Iron oxide-alginate | 1680 | 5 | 0.861 | 0.563 | [106] |
| *Turbinaria-conoides* | 120 | 5 | 1.11 | 5.28 | [107] |
| *Sargassum-polycystum* | n.d. | 5 | 0.98 | 69.4 | [108] |
| Grapefruit peel | 60 | 5 | 1.23 | 5.21 | [109] |
| *Platanus orientalis* leaf | 60 | 4 | 0.206 | 18.1 | [110] |
| $SnO_2$-$TiO_2$ nanocomposites | 60 | 5 | 0.488 | 26.4 | [111] |
| *Sargassum* (sp). | 60 | n.d. | 0.66 | 116 | [112] |
| Alginate magnetite beads | 300 | 4 | 2.03 | 0.87 | [98] |
| Bamboo-charcoal | 480 | 7.2 | 1.38 | 76.5 | [113] |
| Banana peel | 1440 | 5.2 | 0.279 | 361 | [114] |
| Magnetic graphene nanoparticles | 15 | 4 | 0.358 | 35.4 | [115] |
| Purolite-S950 | 180 | 0.2M $HNO_3$ | 0.636 | 12.9 | [91] |
| MOF functionalized sorbent | 40–60 | 7 | 2.08 | 44.4 | [99] |
| *Pseudomonas aeruginosa* | 180 | 5 | 1.00 | n.d. | [116] |
| $A_L$PEI | 40 | 5 | 0.573 | 1.14 | [100] |
| POH-$A_L$PEI | 40 | 5 | 1.61 | 3.85 | [100] |
| PGEG | 40 | 4 | 0.45 | 4.3 | *This work* |
| TBP-PGEG | 25 | 4 | 1.47 | 5.6 | *This work* |

Units: Time: min, $q_{m,L}$: mmol g$^{-1}$; $b_L$: L mmol$^{-1}$.

### 3.2.4. Multi-Component Solutions—Selectivity

Supplementary studies of the experiments were focused on the treatment of La with equimolar polymetallic ions. Prepared equimolar solutions of La, Si, Ca, Mg, and Tb (heavy rare earth elements) were used to investigate the selectivity behavior of the TBP-PGEG in polymetallic solution; most of these ions coexist with La in a leachate solution. These experiments were performed for a $pH_0$ between 1 and 5. Figure 8a reports the selectivity of La against selected elements at different pH values—confirming the preference of sorbent for REEs over the major elements, especially so at higher pH values. The selectivity is in the order Si >> Mg > Ca >> Tb at $pH_{eq}$ = 4.03, with values 49.3, 31.3, 22.9, and 2.4%, respectively. This confirms the high selectivity of the sorbent toward the associated elements and poor selectivity for other REEs. Figure 8b shows the loading capacity of metal ions at different stages which shows individual capacities as follows: 0.085, 0.128, 0.17, 0.53, and 0.65 mmol M g$^{-1}$; here M = Si, Mg, Ca, Tb, and La, respectively.

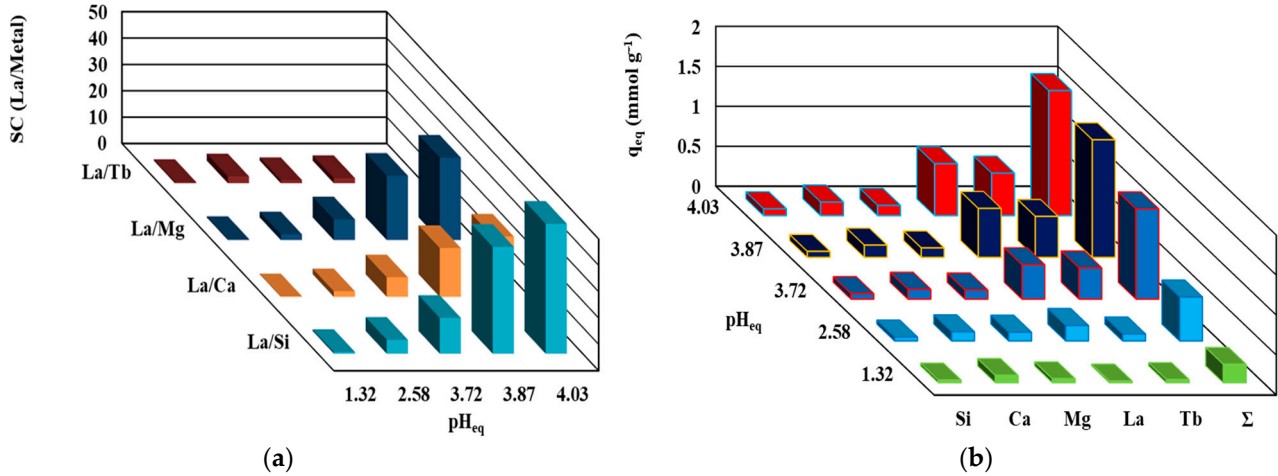

**Figure 8.** The selectivity (**a**) and the loading capacity (**b**) of TBP-PGEG in polymetallic solutions.

### 3.2.5. Desorption and the Sorbent Recycling

The desorption processes of adsorbed La(III) was studied in 20 mL of a 0.5 M HNO₃ solution with 100 mg of each sorbent. The desorption was faster than the adsorption. 20 and 15 min are sufficient for total desorption for PGEG and TBP-PGEG, respectively. These experiments were performed on samples collected from kinetic experiments. Five cycles of sorption were performed. A decrease in the loading capacity for the functionalized sorbent (not exceeding 3.5%) is seen after five cycles, while the desorption efficiency remains around 100%. Table 6 shows the performances of sorption and desorption for TBP-PGEG during the five cycles. S.D. is the standard deviation, R = removal, and De% = desorption in %.

**Table 6.** Sorption desorption performances for the TBP-PGEG sorbent.

| | Sorption | | Desorption | |
|---|---|---|---|---|
| **Cycles** | **Efficiency (%)** | **S.D. (Rem.%)** | **Efficiency (%)** | **S.D. (Des%)** |
| 1 | 85.47 | 1.15 | 99.99 | 0.074 |
| 2 | 84.69 | 1.21 | 99.98 | 0.44 |
| 3 | 82.53 | 0.90 | 99.96 | 0.22 |
| 4 | 81.98 | 0.83 | 100.00 | 0.19 |
| 5 | 81.29 | 0.74 | 99.64 | 0.36 |

### 3.3. Applications

An acidic leaching solution was collected each time after being pretreated for U removal; the concentrations of metal ions in the solution were listed above in Table 1. The treatment was performed via both sorbents at different pH values. The concentration of REEs in the original solution was around 410 mg REE L⁻¹ and decreased to 380 mg REE L⁻¹ after a treatment with quaternary ammonium chloride resin for U(VI) removal with efficiency loss of 7.3%.

Figure 9a,b shows the selectivity performance of La(III) with respect to the metal ions of interest. The figure shows a variety in behaviors. As the pH increases, the sorption of metal ions increases with increasing selectivity as well. Better performance of the TBP-PGEG sorbent towards REEs is seen than that of the PGEG. The selectivity toward metal ions depends on the pH value. In case of PGEG, the maximum reached at $pH_{eq}$ = 5.11 for Mg (SC/5.8), Ca (SC/4.46), Si(SC/2.5), and V (SC/2.3), while at $pH_{eq}$ = 3.28, it reached maximum for Mn(SC/12.12), Fe(SC/7.35), and Al(SC/4.42). In the case of TBP-PGEG, it was noticed mainly at $pH_{eq}$ = 4.79 for Ca (SC/93.7) > Mg (SC/89.9) > Mn(SC/60.1) >> Si(SC/23.7) > V (SC/17.0), while the SC of Al reached maximum at $pH_{eq}$ = 4.09 with value 26.3 and Fe at $pH_{eq}$ = 3.18 with value 48.5.

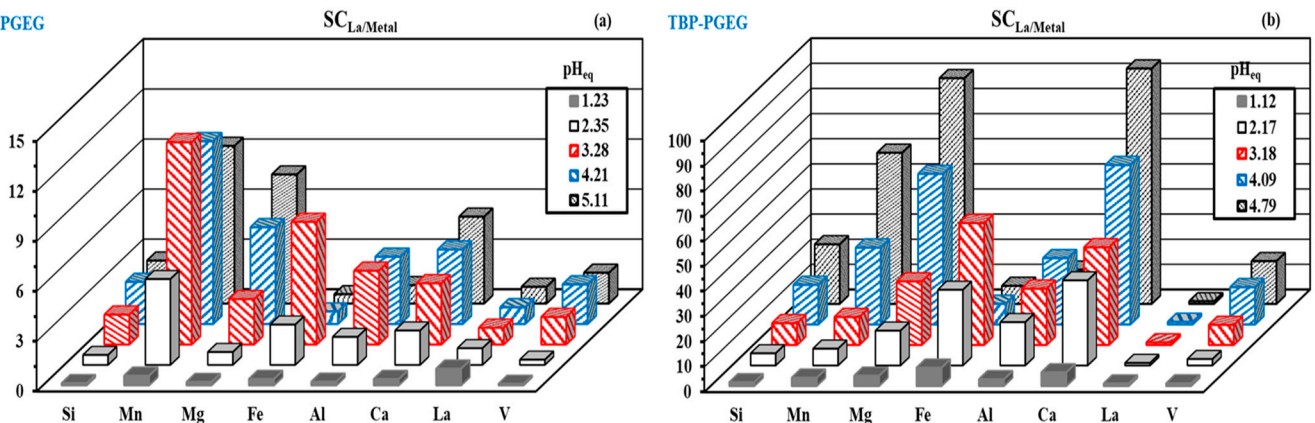

**Figure 9.** Selectivity performance of La vis metal ions for PGEG (**a**) and TBP-PGEG (**b**).

## 4. Conclusions

We have successfully synthesized new hydrogels based on gellan gum and chitosan for pristine polymer (PGEG) and for a functionalized polymer containing phosphonate groups (TBP-PGEG). The new sorbents have been fully characterized by FTIR, TGA, BET, SEM-EDX, and $pH_{PZC}$. The new functionalized sorbent was efficient for the recovering of REEs in a mild acidic solution and also for the recovering of REEs from a leachate solution. The partial deprotonation of functional groups at pH = 4 (amine, hydroxyl, phosphonate, and carboxylate) improves the sorption of REEs. Fast kinetics of the functionalized phosphonate sorbent is seen in the PFORE profile, 25 min rather than 40 min for PGEG. Elution was efficient using 0.5 M $HNO_3$ and the total desorption was achieved in 15 min for both sorbents. The Langmuir and Sips isotherms profiles provide better fits for both sorbents than the Freundlich equation. The maximum sorption capacities are 0.45 and 1.47 mmol La $g^{-1}$ for PGEG and TBP-PGEG, respectively. It shows a high stability for sorption desorption with a slight drop after five cycles. The functionalized sorbent shows a high selectivity toward REEs in treatments of an equimolar synthetic solution (with selectivity 49.3, 31.3, 22.9, and 2.4% for Si, Mg, Ca, and Tb, respectively). The selectivity after the treatment of ore leachate was found to depend on the pH. For PGEG, the maximum selectivity was found at $pH_{eq}$ = 5.11 for Ca and Mg, at $pH_{eq}$ = 4.21 for Si and V, and at $pH_{eq}$ = 3.28 for Mn, Fe, and Al. As for TBP-PGEG, the maximum selectivity occurred at $pH_{eq}$ = 4.79 for Ca, Mg, Mn, Si, and V, while it was at $pH_{eq}$ = 4.09 for Al and at $pH_{eq}$ = 3.18 for Fe.

**Supplementary Materials:** The following supporting information can be downloaded at: https://www.mdpi.com/article/10.3390/su15032843/s1, Table S1: Equations used for modeling sorption isotherms [117–119]; Table S2: Equations used for modeling uptake kinetics [120,121]; Table S3: Constituents of leaching solution before used for U removal at (pH = 0.3); Table S4: FTIR assignments of peaks for PGEG, TBP-PGEG, after loading with La, and after 5 cycles of sorption desorption; Figure S1: Dr-TG analysis of the PGEG and TBP-PGEG sorbents; Figure S2: FTIR of collected spectra focusing on the main peaks [79–81,122–148]; Figure S3: pHpzc values of the PGEG and TBP-PGEG sorbents; Figure S4: Speciation of La at different pH values; Figure S5: PSORE (**a**) and RIDE (**b**) fitting curves of PGEG and TBP-PGEG sorbents; Figure S6: Freundlich equation fitting curves for the sorption isotherms of La(III) ions on PGEG and TBP-PGEG sorbents.

**Author Contributions:** Conceptualization, M.F.H., W.M.A. and D.I.Z.; methodology, M.F.H. and N.A.H.; software, D.I.Z. and Y.W.; validation, M.F.H., Y.W., K.A. and W.B.; formal analysis, M.F.H., D.I.Z., Y.W. and N.A.H.; investigation, K.A. and W.B.; resources, M.F.H., W.M.A., D.I.Z. and N.A.H.; data curation, K.A., W.B. and N.A.H.; writing—original draft preparation, W.B., M.F.H. and D.I.Z.; writing—review and editing, M.F.H.; visualization, K.A., W.B. and N.A.H.; project administration, Y.W.; funding acquisition, Y.W. All authors have read and agreed to the published version of the manuscript.

**Funding:** The National Natural Science Foundation of China for supporting projects [U1967218, and 11975082].

**Institutional Review Board Statement:** Not applicable.

**Informed Consent Statement:** Not applicable.

**Data Availability Statement:** Data/samples can be obtained from Authors on demand.

**Acknowledgments:** Y.W. acknowledges the National Natural Science Foundation of China for supporting the projects [U1967218, and 11975082].

**Conflicts of Interest:** The authors declare no conflict of interest.

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
