# Peer review of "A Phosphonic Functionalized Biopolymer for the Sorption of Lanthanum (III) and Application in the Recovery of Rare Earth Elements"

_sustainability, doi:10.3390/su15032843_

Round 1

Reviewer 1 Report

The publication shows sufficient new and interesting results, especially concerning the Functionaliztion of Biopolymer for the Sorption Study of Lanthanum (III). The language need to be modified. However, there are some drawbacks of the paper due to the inclusion of "Chemical" part in it, which is not well described. Therefore, I suggest that the authors should have a major revision and collation of the article. For more details see the comments bellow.

Q1. In the abstract section the first two sentences need to be modified.

Q2. The Introduction part, the authors must discuss the uses of chetosan in the sorption of Lanthanum.

Q3.  The novelty of the work must be represented in the introduction section.

Q5. Please provide EDX analysis for PGEG and TBP-PGEG sorbents with SEM images in the same discussion and Figs.

Q6 . Page 6  lines 206-207 please use this reference for  water loss (Particulate Science and Technology, 37:2, 207-219, DOI:10.1080/02726351.2017.1362607).

Q6.Page 7 lines 232 and 233 please use this reference for phosphate peaks ((J Adv Pharm Edu Res 2018;8(3):59-67)

Q7. In the desorption study, I think 0.5 M HNO3 is so much, and may be affect on solubility of the sorbent,  so I advance authors to use 0.1 M  HNO3.

Q8. The conclusion part needs modification

Author Response

RESPONSE TO REVIEWERS’ COMMENTS

Red: Specific response to Reviewer comment.

Blue: New statement added to the revised manuscript

We would like to thank the reviewer for the careful reading of the manuscript and meaningful suggestions. We hope the revised version will comply with his/her expectations.

Reviewer #1: Comments to sustainability-2173228:

Review Report Form

Open Review

English language and style

( ) English very difficult to understand/incomprehensible
( ) Extensive editing of English language and style required
( ) Moderate English changes required
(x) English language and style are fine/minor spell check required
( ) I don't feel qualified to judge about the English language and style

Yes

Can be improved

Must be improved

Not applicable

Is the content succinctly described and contextualized with respect to previous and present theoretical background and empirical research (if applicable) on the topic?

(x)

( )

( )

( )

Are all the cited references relevant to the research?

(x)

( )

( )

( )

Are the research design, questions, hypotheses and methods clearly stated?

(x)

( )

( )

( )

Are the arguments and discussion of findings coherent, balanced and compelling?

(x)

( )

( )

( )

For empirical research, are the results clearly presented?

(x)

( )

( )

( )

Is the article adequately referenced?

(x)

( )

( )

( )

Are the conclusions thoroughly supported by the results presented in the article or referenced in secondary literature?

( )

(x)

( )

( )

We would like to thank the reviewer for his globally positive evaluation of our work.

Comments and Suggestions for Authors

The publication shows sufficient new and interesting results, especially concerning the Functionaliztion of Biopolymer for the Sorption Study of Lanthanum (III). The language need to be modified. However, there are some drawbacks of the paper due to the inclusion of "Chemical" part in it, which is not well described. Therefore, I suggest that the authors should have a major revision and collation of the article. For more details see the comments bellow.

Q1. In the abstract section the first two sentences need to be modified.

Thanks for your comment, it was revised and modified and the whole manuscript was revised as well

Q2. The Introduction part, the authors must discuss the uses of chetosan in the sorption of Lanthanum.

Thanks, we support the introduction by this information. Hope it was fine now.

This paragraph was added

Several studies were performed for metal recovery using modified chitosan particles. Crosslinked lanthanum loaded chitosan/ silica gel particles were performed through designing of LaCl3·7H2O, chitosan and silica gel composite, which crosslinked using glutaraldehyde crosslinker to yield (LaSiCS), it was considered as an excellent sorbent for chromium removal[63].  Another sorbent was synthesized and explored for removing of lanthanum and lead ions from water using manganese dioxide chitosan composite (crosslinked by formalde-hyde), it shows a high affinity and rapidness toward these ions[64]. Designing of sorbents based on chitosan and polyvinylpyrrolidone (PVP) particles were designed for separating of La(III) from (La(III), Ce(III), and Sm(III)). Tartaric acid with concentration of 0.25 M, was used as an eluent for lanthanide from Ln(III)IP-CS/PVP particles, which considered to have a selective extraction performance[65].

Q3.  The novelty of the work must be represented in the introduction section.

Thanks for alerting, we add some sentences represent the novelty of the work.

This paragraph was added

The functionalized sorbent (TBP-PGEG) shows as high loading capacity, with fast kinetics than the pristine sorbent (PGEG) through test toward La(III). The loading capacities ranged 0.226 and 0.78 mmol La g-1 for the PGEG and TBP-PGEG respectively, which exceeding by fold. Other advantage for TBP-PGEG concerning to the total sorption, which was achieved through 25 min comparing to 40 min for the PGEG sorbent. The functionalized sorbent shows high stability against acidic solution (0.5 M HNO3) through sorption desorption cycles.  It is noteworthy that a high affinity was noticed toward REEs from acidic solution that make it as a promising tool for recovering of it from leachate solution.

Q5. Please provide EDX analysis for PGEG and TBP-PGEG sorbents with SEM images in the same discussion and Figs.

Thanks, we merge the SEM and EDX analysis of the PGEG and TBP-PGEG sorbents in one figure.

Q6 . Page 6  lines 206-207 please use this reference for  water loss (Particulate Science and Technology, 37:2, 207-219, DOI:10.1080/02726351.2017.1362607).

Thanks for alerting, it was added

Q6.Page 7 lines 232 and 233 please use this reference for phosphate peaks ((J Adv Pharm Edu Res 2018;8(3):59-67)

Thanks for alerting, it was added

Q7. In the desorption study, I think 0.5 M HNO3 is so much, and may be affect on solubility of the sorbent,  so I advance authors to use 0.1 M  HNO3.

Thanks for alerting, Actually the sorbent exhibited a high stability during sorption desorption cycles and the chemical composition was not broken. This was verifying through the FTIR after 5 cycles of sorption desorption. We thanks the reviewer for the suggestion and we will take it in our consideration in the future work.

Q8. The conclusion part needs modification

Thanks for mention, it was revised and modified.

Reviewer 2 Report

The manuscript (sustainability-2173228) entitled "Phosphonic Functionalized Biopolymer for the Sorption Study of Lanthanum (III) and Application for the Recovery of Rare Earth Elements" deals with the synthesis of a hydrogel, based on gellan gum, chitosan and grafted by phosphonate, its characterization, and application as a sorbent for lanthanum (III) adsorption from aqueous solutions.

This work could be of interest to the scientific community. However, a minor revision of the manuscript is needed before it can be accepted for publication in this Journal.

Some specific comments are given below.

Those words that are already in the title should be removed from the keywords. Keywords serve the purpose of indexing and there is no need to repeat those words that are already in the title.

In the case of experimental results, the number of significant figures should be reduced to two or three in accordance with the precision of the measurement. Too many digits give a false impression of precision and those extra digits are inaccurate and useless.

The literature should be updated, that is, research published in the last two years in the field of recovery of rare earth elements from aqueous solutions.

The figures should be revised so that the quality is greatly improved.

Attention should be paid to the abbreviations in the manuscript. Abbreviations should be described when they first appear in the manuscript.

There are a large number of grammatical errors in the work that need to be corrected.

Author Response

RESPONSE TO REVIEWERS’ COMMENTS

Red: Specific response to Reviewer comment.

Blue: New statement added to the revised manuscript

We would like to thank the reviewer for the careful reading of the manuscript and meaningful suggestions. We hope the revised version will comply with his/her expectations.

Reviewer #3: Comments to sustainability-2173228:

Review Report Form

Open Review

English language and style

( ) English very difficult to understand/incomprehensible
( ) Extensive editing of English language and style required
(x) Moderate English changes required
( ) English language and style are fine/minor spell check required
( ) I don't feel qualified to judge about the English language and style

Yes

Can be improved

Must be improved

Not applicable

Is the content succinctly described and contextualized with respect to previous and present theoretical background and empirical research (if applicable) on the topic?

( )

(x)

( )

( )

Are all the cited references relevant to the research?

( )

(x)

( )

( )

Are the research design, questions, hypotheses and methods clearly stated?

( )

(x)

( )

( )

Are the arguments and discussion of findings coherent, balanced and compelling?

( )

(x)

( )

( )

For empirical research, are the results clearly presented?

( )

(x)

( )

( )

Is the article adequately referenced?

( )

(x)

( )

( )

Are the conclusions thoroughly supported by the results presented in the article or referenced in secondary literature?

( )

(x)

( )

( )

Comments and Suggestions for Authors

The manuscript (sustainability-2173228) entitled "Phosphonic Functionalized Biopolymer for the Sorption Study of Lanthanum (III) and Application for the Recovery of Rare Earth Elements" deals with the synthesis of a hydrogel, based on gellan gum, chitosan and grafted by phosphonate, its characterization, and application as a sorbent for lanthanum (III) adsorption from aqueous solutions.

 This work could be of interest to the scientific community. However, a minor revision of the manuscript is needed before it can be accepted for publication in this Journal.

 Some specific comments are given below.

 Those words that are already in the title should be removed from the keywords. Keywords serve the purpose of indexing and there is no need to repeat those words that are already in the title.

Thanks for mention, it was revised and modified.

The new Keywords are

Biopolymer, phosphorylated sorbent, uptake kinetics, sorption isotherms, rare earth elements

In the case of experimental results, the number of significant figures should be reduced to two or three in accordance with the precision of the measurement. Too many digits give a false impression of precision and those extra digits are inaccurate and useless.

Thanks for alerting. We already add the essential for the reader and built another file which have unessential as the unfitted for isotherms and kinetics. we try to reduce as the reviewer demand and hope the revised version receive his satisfactions. 

The literature should be updated, that is, research published in the last two years in the field of recovery of rare earth elements from aqueous solutions.

Thanks for your comments. Yes, there are a lot of work concerning with sorption of La(III) but we choose the most fitted composites related to our material (biopolymer) and also the type of tested solution, this is to make a meaningful of the comparison.

 The figures should be revised so that the quality is greatly improved.

Thanks, we revised it and improve the quality

Attention should be paid to the abbreviations in the manuscript. Abbreviations should be described when they first appear in the manuscript.

Thanks for the meaningful comment and revised it and try to follow the rules

 There are a large number of grammatical errors in the work that need to be corrected.

Thank you for alerting. We revised the whole manuscript and we carefully checked the Editing (typing and grammar). We hope the revised version is meaningful and making the manuscript readable.

Round 2

Reviewer 1 Report

The authors made all requirements and revised it carefully, I think the manuscript titled "Phosphonic Functionalized Biopolymer for the Sorption Study of  Lanthanum (III) and Application for the Recovery of Rare Earth  Element" is suitable for publication in its present form in Sustainability.

Author Response

We would like to thanks the reviewer for his/her global recommendation, we revised the manuscript carefully by native English speaker Prof. Witold Brostow from the University of North Texas, he is a coauthor on this work